# Hydrophilic Linear Peptide with Histidine and Lysine Residues as a Key Factor Affecting Antifungal Activity

**DOI:** 10.3390/ijms19123781

**Published:** 2018-11-28

**Authors:** Seong-Cheol Park, Jin-Young Kim, Eun-Ji Kim, Gang-Won Cheong, Yongjae Lee, Wonkyun Choi, Jung Ro Lee, Mi-Kyeong Jang

**Affiliations:** 1Department of Polymer Science and Engineering, Sunchon National University, Suncheon, Jeonnam 57922, Korea; schpark9@gnu.ac.kr (S.-C.P.); jyfrog@hanmail.net (J.-Y.K.); kej2671@sunchon.ac.kr (E.-J.K.); 2Division of Applied Life Sciences and Research Institute of Natural Science, Gyeongsang National University, Jinju, Gyeongnam 52828, Korea; gwcheong@gnu.ac.kr; 3Department of Nutrition and Food Science, Texas A&M University, College Station, TX 77843, USA; youngjaelee@exchange.tamu.edu; 4National Institute of Ecology (NIE), Seocheon, Choongnam 33657, Korea; wonkyun@nie.re.kr; 5The Research Institute for Sanitation and Environment of Coastal Areas, Sunchon National University, Suncheon, Jeonnam 57922, Korea

**Keywords:** antimicrobial peptide, drug-resistance, antifungal activity, reactive oxygen species, apoptosis

## Abstract

Increases in the numbers of immunocompromised patients and the emergence of drug-resistance fungal pathogens have led to the need for new, safe, efficacious antifungal agents. In this study, we designed a histidine-lysine-lysine (HKK) motif and synthesized six HKK peptides with repetitions of the motif. These peptides showed length-dependent antifungal activity against drug-susceptible and drug-resistant fungal pathogens via membranolytic or non-membranolytic action. None of the peptides were cytotoxic to rat erythrocytes or NIH3T3 mouse embryonic fibroblasts. Short-length peptides were directly translocated in fungal cytosol and reacted with mitochondria, resulting in apoptosis. Membrane-permeabilizing activity occurred in the presence of long peptides, and peptides were able to transfer to the cytosol and induce reactive oxygen species. Our results suggest that peptides composed only of cationic amino acids may be good candidates as antifungal agents.

## 1. Introduction

The incidence and diversity of invasive fungal infections are growing, and fungal pathogens are a global threat to public health. This is particularly so in immunocompromised patients with human immunodeficiency virus/acquired immunodeficiency syndrome or cancer [1,2]. Unfortunately, the existing antibiotic agents that act against fungi, such as flucytosine and azole agents, are ineffective because of the appearance of multidrug-resistant strains. Thus, there is an urgent need for the discovery and development of novel antifungal agents [3]. Promising candidates include antimicrobial peptides (AMPs) isolated from diverse sources such as plants, bacteria, fungi, and animals [4]. AMPs have a selective targeting ability in pathogens and multiple modes of action, thus offering less opportunity for the development of resistance [5]. 

AMPs are the first defense molecules in the innate immune response among all classes of life. They have the following common characteristics: relatively short length (10–50 amino acids), positive charge, amphipathic structure, and electrostatic and hydrophobic interactions with the microbial cell wall or membrane [6,7,8,9]. AMPs inhibit or kill a broad spectrum of invading pathogens including bacteria, fungi, parasites, and enveloped viruses, as well as cancer cells [5,9]. Notably, resistance to AMPs is very rare because AMPs kill microbes mostly by membrane disruption mechanisms, unlike the metabolic inhibitory mechanisms of conventional antibiotics, which can easily induce drug resistance. Several researchers have designed synthetic AMPs via template-based, biophysical, and virtual screening studies [10], rather than by natural purification. In effective AMP design, the most important consideration is cell-selectivity because highly cationic peptides may attack mammalian cells rather than pathogenic microbes. In particular, antifungal peptide design is relatively difficult because fungal cell membranes have mainly zwitterionic phospholipids and sterols, similar to mammalian cell membranes. 

To date, the design of antifungal peptides has been limited to amphipathic α-helical or β-strand structures composed of hydrophobic and cationic amino acids. Accordingly, in this study we aimed to develop a novel antifungal agent based on an antifungal histone-lysine-lysine (HKK) motif, which consisted of only two hydrophilic amino acids, histidine and lysine, and showed antifungal activity. Furthermore, we investigated the antifungal mechanism of the peptides.

## 2. Results and Discussion

### 2.1. Design, Antifungal Activity, and Cytotoxic Effects of Peptides

The amphipathic structure is a common physicochemical characteristic of membranolytic AMPs and has been shown to be important for enabling binding, insertion, and disintegration into the microbial membrane. Moreover, although ultra-short linear AMPs without amphipathicity have also been identified and designed [11,12], these peptides have both hydrophobic and hydrophilic amino acids. In this study, we designed peptides with single, double, triple, quadruple, sextuple, or octuple repetitions of the HKK motif composed of only hydrophilic amino acids. The p*K*a value of the imidazole side chain in histidine is approximately 6.0, unlike the p*K*a of 10.5 in the amino side chain of lysine. Although the pH values of cellular compartments, body fluids, and organs are different, physiological pH in normal cells, tissues, and body fluids is near 7.4, which is similar to that in blood. The environmental pH of tissues and cells infected by fungal pathogens can be altered from 4 to 7.5 within a few hours; in particular, *Candida albicans* induces an acidified environment to express virulence and pathogenesis [13,14]. Cationic peptides containing histidine residues are fully positively charged when the imidazole side groups of histidine are protonated under an acidic environment [15]. We assumed that the lower isoelectric point of the histidine-containing peptide, compared with that of the peptide with lysine only, would contribute to the cell selectivity between normal cells and fungal cells around infected tissues when the designed antifungal peptides were applied in the body.

To determine the minimum length of the designed peptides that influenced antifungal activity, we evaluated the minimum inhibitory concentrations (MICs) of each peptide against nine filamentous and seven yeast fungi, including drug-resistant *C. albicans* strains; the results are summarized in Table 1. HKK peptides inhibited the growth of all tested fungi in a length-dependent manner. Interestingly, fluconazole was inactive against drug-resistant *C. albicans* strains (Culture Collection of Antimicrobial Resistant Microbes; CCARM 14001, 14004, 14007, and 14020) at 256 µM (data not shown), whereas most of the HKK peptides exhibited low MICs ranging from 8 to 64 µM, except for (HKK)_1_ peptide. 

Hemolysis and cell survival were measured to assess the toxicity of HKK peptides to eukaryotic cells. Although melittin showed 100% hemolysis and cell death at 3.13 µM against rat erythrocytes and HaCaT cells, respectively, all HKK peptides were nonhemolytic and noncytotoxic within the tested concentration ranges (Figure 1B,C). In general, fungal cell membranes have negatively charged phosphatidylinositol and phosphatidic acids, whereas mammalian cell membranes lack these negatively charged lipids. Thus, our results suggested that HKK peptides may selectively target or kill fungal pathogens when applied in the clinical setting.

### 2.2. Localization and Membrane-Permeable Effects of HKK Peptides

To investigate cellular compartments through which HKK peptides act in *C. albicans* cells, localization of FNR675-labeled peptides was observed by confocal laser-scanning microscopy (CLSM). As shown in Figure 2A, all tested peptides penetrated into the cells and accumulated in the cytosol. In particular, FNR675-labeled (HKK)_6_ and (HKK)_8_ peptides were clustered around cytosolic parts or organelles, and the morphologies of cells treated for the same length of time were different. For example, *C. albicans* cells treated with (HKK)_6_ peptide had swelled, whereas cells treated with (HKK)_8_ had shrunk. These results suggested that (HKK)_6_ and (HKK)_8_ peptides may have two-step mechanisms, i.e., cell wall and intracellular damage. Furthermore, the (HKK)_8_ peptide showed faster cell binding, penetration, and intracellular damage effects than the (HKK)_6_ peptide, for which membrane damage was not observed with CLSM. Therefore, we next investigated the membrane-permeable effects of the peptide using SYTOX Green uptake assays. In these assays, the fluorescent nuclear dye SYTOX Green is impermeable to live cells and can penetrate cells only when the peptide disrupts the cell membrane; the dye that is taken up into the cells then emits green fluorescence. 

As shown in Figure 2B, almost all cells treated with melittin, a membranolytic peptide, were green-shifted by SYTOX Green uptake. In contrast, histatin 5, a penetrating peptide, did not induce membrane damage. Green fluorescence in fungal cells treated with (HKK)_1_, (HKK)_2_, (HKK)_3_, and (HKK)_4_ was not detected by flow cytometry, whereas (HKK)_6_ and (HKK)_8_ permitted SYTOX Green uptake into the cytosol, indicating that both peptides destabilized the fungal cell membrane before they were internalized into the cytosol. 

To further investigate the membrane-permeable effects of peptides, phosphatidylcholine (PC)/phosphatidylethanolamine (PE)/phosphatidylinositol (PI)/ergosterol (5:4:1:2, *w*/*w*/*w*/*w*) and PC/cholesterol (Ch)/sphingomyelin (SM; 1:1:1, *w*/*w*/*w*) vesicles were used as artificial fungal and mammalian model membranes, respectively. As shown in Figure 3A, melittin induced a marked concentration-dependent calcein release from the fungal model membrane. Calcein was not released from liposomes with (HKK)_1_, (HKK)_2_, (HKK)_3_, and (HKK)_4_ peptides; whereas, (HKK)_6_ and (HKK)_8_ peptides resulted in 12.8% and 21.5% calcein leakage, respectively, at a molar ratio of 0.05 (lipid/peptide). These results were consistent with data from the SYTOX Green uptake. Meaningful calcein leakage was not detected in the presence of any of the HKK peptides in mammalian artificial liposomes (Figure 3B). However, melittin showed concentration-dependent calcein leakage, suggesting that HKK peptides were nontoxic and that melittin may cause severe toxicity in clinical applications, consistent with the findings presented in Figure 1B,C. 

### 2.3. Reactive Oxygen Species (ROS) Production via Mitochondrial Damage

Lipids in the mitochondrial membrane are composed of phospholipids, sterols, and sphingolipids. Among phospholipids, the contents of cardiolipin, an anionic phospholipid, ranged from 10% to 15% higher than in other subcellular organelles [16]. Accordingly, we assumed that HKK peptides may induce ROS production via mitochondrial damage if their linear structure enabled them to penetrate easily across the fungal membrane. ROS containing oxygen-centered radicals (i.e., OH, ROO, RO, and O_2_-) and non-radicals (^1^O_2_ and H_2_O_2_) can be generated by exposure to antibiotics and AMPs. The superoxide anion is produced when the mitochondria of fungal cells are damaged by complete depolarization or disruption of the mitochondrial membrane, and are converted to H_2_O_2_ by superoxide dismutase, followed by production of the highly reactive hydroxyl radical via Haber-Weiss and Fenton reactions [17,18]. Intracellular ROS accumulation can induce fungal membrane damage by lipid peroxidation and fungal apoptosis by activation of metacaspase [19]. 

To evaluate ROS generation by the HKK peptides in fungal cells, 2′,7′-dichlorofluorescein diacetate (DCFH-DA), which is oxidized by hydrogen peroxide, hydroxyl radicals, and peroxynitrite to yield DCF via peroxidase was used to treat cells after incubation with each peptide. As shown in Figure 4A, high levels of ROS generation from *C. albicans* cells were detected in the presence of 10 mM H_2_O_2_. In the presence of HKK peptides, the accumulation of excessive ROS was significantly increased as the number of repeats of the HKK motif increased. 

In order to investigate intracellular targeting or the mechanisms of ROS generation, we measured the generation of mitochondrial superoxide using a MitoSOX Red fluoroprobe in peptide-treated *C. albicans* cells (Figure 4B). Melittin did not generate mitochondrial superoxide (SOX) because it formed toroidal pores onto the fungal membrane. Mitochondrial SOX was not detected in cells with (HKK)_1_ and (HKK)_2_ peptides, indicating nontargeting mitochondria; whereas, (HKK)_3_, (HKK)_4_, (HKK)_6_, and (HKK)_8_ peptides showed significant mitochondrial SOX generation, similar to histatin 5. The results suggested that (HKK)_6_ and (HKK)_8_ peptides may penetrate the cells after membrane destabilization or via transient pores. 

### 2.4. Apoptosis Induction

Similar to other eukaryotes, apoptosis in fungi proceeds through mitochondrial dysfunction, ROS production, cytochrome *c* release, metacaspase activation, phosphatidylserine externalization, DNA and nuclear fragmentation, and chromatin condensation. In addition, apoptotic cells exhibit cell shrinkage and membrane blebbing [20,21]. Antibiotic molecules, including amphotericin B [22], miconazole [23], itraconazole [24], curcumin [25], resveratrol [26], thymol [27], and honokiol [28], and antifungal peptides and proteins, including coprisin [29], rice defensin [30], arabidopsis profilin [31], SGT1 [32] and UST [33], and scolopendin 1 [34], inhibit fungal growth by inducing apoptosis.

Therefore, we next measured the release of cytochrome *c* from the mitochondria into the cytosol to investigate the induction of the apoptotic cascade via mitochondrial dysfunction. In the presence of HKK peptides, the relative cytochrome *c* levels in the cytosol increased in a peptide length-dependent manner; however, histatin 5 induced only minor alterations in cytochrome *c* levels (Figure 5A). These results indicated that HKK peptides may cause apoptosis via mitochondrial dysfunction. In addition, terminal deoxynucleotidyl transferase dUTP nick end labeling (TUNEL) assays were performed to further evaluate induction of apoptosis. 3′-Hydroxyl termini in double-stranded DNA breaks during apoptosis, labeled using terminal deoxyribonucleotidyl transferase, were observed as green fluorescent signals. As shown in Figure 5B, cells treated with (HKK)_6_ and (HKK)_8_ peptides displayed DNA breaks in TUNEL assays. The results shown in Figure 4 and Figure 5 suggested that HKK peptides inhibited the growth of fungal cells via mitochondrial ROS generation and induction of apoptosis. 

### 2.5. Morphological Alterations in *C. albicans* Cells with HKK Peptides

Next, we observed the morphological and cell surface changes in *C. albicans* cells treated with HKK peptides with scanning electron microscopy. The control cells exhibited a smooth surface without cell debris or unusual aggregates (Figure 6a). However, cells with histatin 5 were wrinkled and shrunken (Figure 6b), and melittin induced damage and pores in the cell walls (Figure 6c). Small blebs were observed on the surfaces of cells treated with (HKK)_1_ or (HKK)_2_ peptides (Figure 6d,e), and the surfaces of other HKK peptide-treated cells showed more roughening and blebbing (Figure 6f–h) than in cells treated with melittin. The antifungal mechanism of melittin has been reported to occur via formation of toroidal pores and membranolytic action in the fungal membrane [35,36,37]. The results indicated that the mode of action of HKK peptides differed from that of melittin. 

## 3. Materials and Methods

### 3.1. Materials

PC, PE, and PI were obtained from Avanti Polar Lipids, Inc. (Alabaster, AL, USA). 9-Fluorenylmethoxycarbonyl (Fmoc) amino acids and Oxyma pure were obtained from CEM Co. (Matthews, NC, USA). SM, Ch, ergosterol, trifluoroacetic acid (TFA), triisopropylsilane (TIS), diisopropylcarbodiimide (DIC), and phenol were purchased from Sigma-Aldrich Co. (St. Louis, MO, USA). DCFH-DA, MitoSOX Red, and SYTOX Green were obtained from Molecular Probes Inc. (Eugene, OR, USA). FNR675 *N*-hydroxysuccinimide ester was purchased from BioActs (Incheon, Korea). All other reagents were of analytical grade.

### 3.2. Peptide Synthesis

All peptides, including (HKK)_1_ (HKK-NH_2_), (HKK)_2_ (HKKHKK-NH_2_), (HKK)_3_ (HKKHKKHKK-NH_2_), (HKK)_4_ (HKKHKKHKKHKK-NH_2_), (HKK)_6_ (HKKHKKHKKHKKHKKHKK-NH_2_), (HKK)_8_ (HKKHKKHKKHKKHKKHKKHKKHKK-NH_2_), melittin (GIGAVLKVLTTGLPA-LISWIKRKRQQ-NH_2_), were synthesized by solid-phase methods with Fmoc-protected amino acids on a Liberty microwave peptide synthesizer (CEM Co., Matthews, NC, USA). Rink amide 4-methylbenzhydrylamine resin (Novabiochem; 0.55 mmol/g) was used to create the amidated peptides. The coupling step and Fmoc deprotection were achieved using microwave heating in the presence of DIC/Oxyma and 20% piperidine in dimethylformamide, respectively. The peptides were cleaved from the resins using a cleavage cocktail (TFA/H_2_O/phenol/TIS 85/5/5/5), followed by precipitation and extraction with ether. After synthesis of the peptides, the crude peptides were purified by a ZORBAX PrepHT Eclipse C_18_ preparative column (21.2 × 150 mm, 5-μm) on a Waters preparative high-performance liquid chromatography system, using a 0%–60% acetonitrile gradient in water with 0.1% trifluoroacetic acid. The purity of the isolated peptides was more than 99.5%, and their molecular masses were measured using a matrix-assisted laser desorption ionization mass spectrometer (MALDI II; Kratos Analytical Ltd., Manchester, UK). 

### 3.3. Antifungal Assay

*Aspergillus flavus* (KCTC 6905), *C. gloeosporioides* (KCTC 6169), *C. albicans* (KCTC 7270), *C. albicans* (CCARM 14001, fluconazole-resistant strain), *C. albicans* (CCARM 14004, fluconazole-resistant strain), *C. albicans* (CCARM 14007, amphotericin B-resistant strain), *C. albicans* (CCARM 14020, fluconazole-resistant strain), *C. krusei* (CCARM 14017), *C. parapsilosis* (CCARM 14016), *F. graminearum* (KCTC 16656), *F. moniliforme* (KCTC 6149), *F. oxysporum* (KCTC 16909), *F. solani* (KCTC 6326), *Tricophyton rubrum* (KCTC 6345), *Trichoderma virens* (KCTC 16924), and *Trichoderma viride* (KCTC 16992) were obtained from the Korea Collection for Type Cultures (KCTC, Jeongup-si, Jeollabuk-do, Korea) and the Culture Collection of Antimicrobial Resistant Microbes (CCARM, Seoul Women’s University, Seoul, Korea).

Spores of 3-day-old fungal cultures grown on potato dextrose (PD; Difco, Sparks, MD, USA) agar plates were collected with 0.08% Triton X-100, and yeast cells were subcultured in yeast extract peptone dextrose (YPD; Difco). Fungal cells were then adjusted to a density of 2 × 10^4^ spores/mL in 4-morpholineethanesulfonic acid (MES) buffer (pH 6.0) containing 20% PD for molds or YPD for yeast. Fungal cells were added to two-fold serially diluted peptides in 96-well plates. After 48 h of incubation at 28 °C, mycelium growth and cell proliferation were observed on an inverted light microscope. The MIC values were taken as the lowest concentration of samples that completely inhibited the visible growth or proliferation of the cells. All assays were performed in triplicate [38].

### 3.4. Hemolysis and Cytotoxicity

Fresh rat red blood cells (rRBCs) were washed with phosphate-buffered saline (PBS) until the supernatant was clear. The rRBC cells were added to a final concentration of 8% (*v*/*v*) in two-fold serially diluted peptide solution. The samples were incubated with mild agitation for 1 h at 37 °C. After centrifugation for 10 min at 800× *g*, the absorbance of the supernatant was measured at 414 nm. Each measurement was made in triplicate, and the percent hemolysis was calculated using the followed equation: % hemolysis = ([Abs_414_ in the peptide solution − Abs_414_ in PBS]/[Abs_414_ in 0.1% Triton-X100 − Abs_414_ in PBS]) × 100(1)

In this equation, 100% hemolysis was defined as the absorbance of rRBCs containing 1% Triton X-100, and zero hemolysis consisted of rRBCs alone in PBS [39]. 

Cytotoxicity was performed using 3-(4,5-dimethyl-2-thiazolyl)-2,5-diphenyl-2H-tetrazolium bromide (MTT) assays. NIH/3T3 mouse fibroblasts were cultured in Dulbecco’s modified Eagle medium supplemented with antibiotics (100 U/mL penicillin, 100 µg/mL streptomycin) and 10% fetal bovine serum at 37 °C in a humidified chamber in an atmosphere containing 5% CO_2_. In total, 5 × 10^3^ cells/well was seeded onto a 96-well plate and incubated overnight, after which peptides were added to the plate, and the plates were further incubated for 24 h. An aliquot of 5 mg/mL MTT was added to each well, and samples were incubated for an additional 4 h. The supernatants were aspirated, and 200 μL dimethyl sulfoxide was added to dissolve any remaining precipitate. Absorbance was then measured at a wavelength of 570 nm using a microtiter SpectraMax M5 reader (Molecular Devices, Sunnyvale, CA, USA) [39]. 

### 3.5. CLSM

The cellular distributions of FNR675-labeled peptides in fungal cells were analyzed with CLSM. After 1 h incubation of *C. albicans* cells with FNR 675-labeled peptides at 28 °C, the washed cells were spotted onto coverglass with mounting solution (50% glycerol and 0.1% *n*-propyl-gallate) and observed with CLSM (LSM 510 META; Gottingen, Germany). Zeiss LSM image software was used for image acquisition and analysis.

### 3.6. SYTOX Green Uptake

*C. albicans* cells were pre-incubated with 0.5 μM SYTOX Green for 15 min in the dark, incubated with the indicated peptides at MICs for 30 min, and then analyzed using a CytoFLEX flow cytometer (Beckman Coulter Inc., Brea, CA, USA). 

### 3.7. Calcein Leakage

The dried lipids were prehydrated with dye buffer solution (80 mM calcein, 10 mM MES, 50 mM NaCl, pH 6.2), vortexed for 1 min, and incubated for 30 min at 50 °C. The suspension was freeze-thawed in liquid nitrogen for nine cycles to generate large unilamellar vesicles (LUVs) and extruded 30 times through polycarbonate filters (two stacked 0.2-μm pore-size filters) using an *Avanti* Mini-Extruder (Avanti Polar Lipids Inc., Alabaster, AL, USA). Calcein-entrapped vesicles were separated from free calcein by gel filtration chromatography on a Sephadex G-50 column. Entrapped LUVs in suspensions containing 10 μM lipids were incubated with various concentrations of the peptide. The fluorescence of the released calcein was assessed with a microtiter SpectraMax M5 reader at an excitation wavelength of 480 nm and an emission wavelength of 520 nm. Complete (100%) release was achieved via the addition of Triton X-100 to a final concentration of 0.1% (*v*/*v*). The apparent percentage of calcein release was calculated in accordance with the following equation:

Release (%) = 100 × (*F* − *F*_0_)/(*F*_t_ − *F*_0_)
(2)
where *F* and *F*_t_ represent the fluorescence intensity after the addition of the peptides and Triton X-100, respectively, and *F*_0_ represents the fluorescence of the intact vesicles.

### 3.8. Measurement of Intracellular ROS and Mitochondrial SOX

*C. albicans* cells pretreated with histatin 5, melittin, or HKK peptides at MICs for 6 h were incubated with 10 μM DCFH-DA for 20 min in the dark. DCF fluorescence was measured using 5000 events/sample following excitation with a 488-nm wavelength laser and reading through a 525/40 biopass filter using a CytoFLEX flow cytometer.

Mitochondrial SOX was measured in *C. alibicans* cells using a MitoSOX Red probe. Fungal cells were pre-incubated with histatin 5, melittin, or HKK peptides at MICs for 6 h, and fluorescent staining was performed following the manufacturer’s protocol for MitoSOX Red. The cells emitting red fluorescence were counted using a CytoFLEX flow cytometer. 

### 3.9. Release of Cytochrome c

To measure the leakage of cytochrome *c* from mitochondria to the cytosol, mitochondria in cells incubated with histatin 5, (HKK)_2_, (HKK)_4_, (HKK)_6_, and (HKK)_8_ at MICs for 6 h were prepared using a Mitochondrial Yeast Isolation Kit (Abcam, Cambridge, MA, USA). Next, 100% ascorbic acid, a reducing agent for cytochrome *c*, was added, and cytochrome *c* levels were measured at 550 nm. 

### 3.10. TUNEL Assay

After pre-incubation of *C. albicans* cells with histatin 5 or HKK peptides for 6 h, TUNEL assays were performed with the prepared protoplasts using a DeadEnd fluorometric TUNEL system (Promega, Fitchburg, WI, USA) according to the manufacturer’s instructions. The nuclei were stained with DAPI. 

### 3.11. SEM 

Precultured *C. albicans* cells (1 × 107 cells/mL) were incubated with histatin 5, melittin, or HKK peptides at the MICs for 1 h. Cell suspensions were dropped on poly-l-lysine coated glass slides, and the attached cells were fixed with 2% glutaraldehyde (*v*/*v*) in 0.2 M HEPES buffer (pH 7.4) overnight at 4 °C, followed by dehydration through a graded ethanol series (50%/75%/100%) and drying in a critical point drier under CO_2_. Gold-coated samples were observed by field emission SEM (JSM-7100F; JEOL, Ltd., Tokyo, Japan).

### 3.12. Statistical Analysis

The mean values of at least four independent determinations ± SD and the *p*-values (Student’s *t*-test) were calculated using Excel software. 

## 4. Conclusions

In summary, we designed novel antifungal peptides with variable molecular masses based on the HKK motif and evaluated the antifungal activity of these HKK peptides at their MICs against 16 phytopathogenic and human-pathogenic fungal strains, including drug-resistant *C. albicans* strains. None of the HKK peptides induced significant hemolysis or cytotoxicity against rRBCs or HaCaT cells, respectively. Based on our mechanistic study, we suggest that randomly coiled HKK peptides may easily penetrate across the fungal cell wall and membrane and bind to the mitochondrial membrane, due to strong electrostatic interactions. The bound peptides induced mitochondrial damage, mitochondrial ROS, and cell apoptosis. Our research supported the use of only cationic amino acids, particularly histidine and lysine, as new design components for antifungal peptides. Further studies are needed to evaluate the in vivo activity of HKK peptides against fungal-infected animal models.

## Figures and Tables

**Figure 1 ijms-19-03781-f001:**
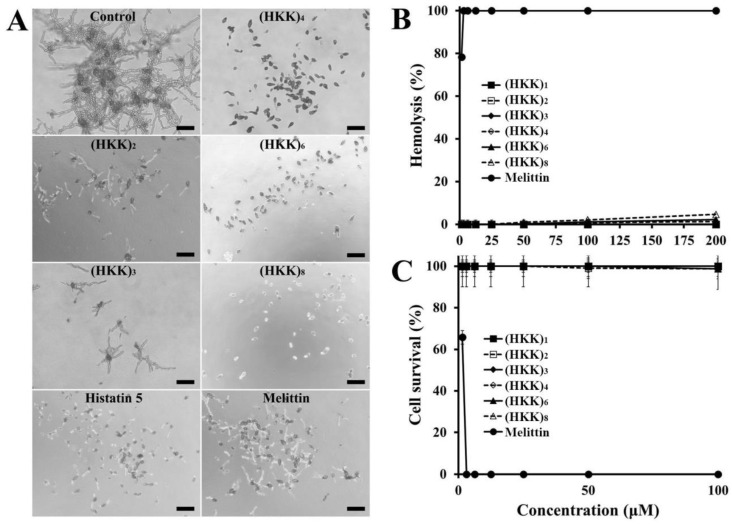
Growth-inhibitory and cytotoxic effects of the peptides. (**A**) After 2 days of incubation with HKK peptides, histatin 5 or melittin, the growth of *Fusarium solani* cells was visualized under an inverted microscope. Control (untreated), (HKK)_2_ (32 µM), (HKK)_3_ (16 µM), (HKK)_4_ (4 µM), (HKK)_5_ (2 µM), (HKK)_6_ (1 µM), histatin 5 (32 µM), and melittin (8 µM) samples were evaluated. Bar: 40 µm. (**B**) After 1 h of incubation with the indicated peptides in 8% rat red blood cells (rRBCs), hemoglobin release was measured. (**C**) After 24 h of incubation with the peptides shown in HaCaT cells, cell proliferation was evaluated by MTT assay.

**Figure 2 ijms-19-03781-f002:**
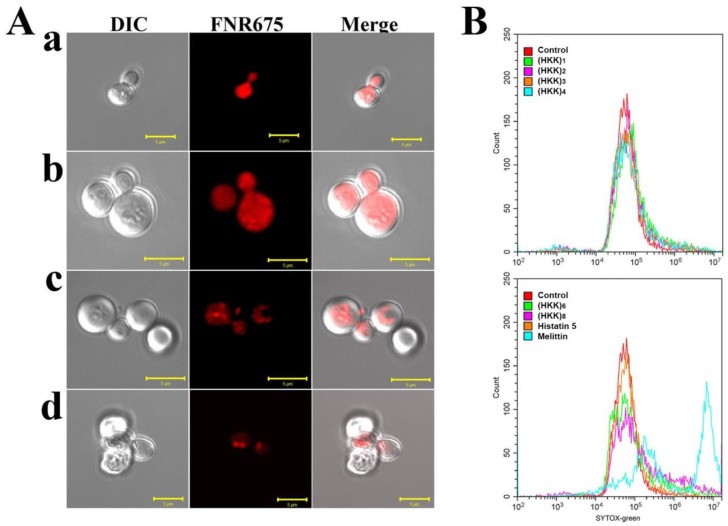
Cellular distribution and membrane-permeable effects of peptides in *C. albicans* cells. (**A**) After 1 h of incubation of *C. albicans* cells with FNR675-labeled (HKK)_2_ (a), (HKK)_4_ (b), (HKK)_6_ (c), and (HKK)_8_ (d) peptides, the washed cells were observed under confocal laser-scanning microscopy (CLSM). (**B**) SYTOX Green-pretreated *C. albicans* cells in the presence of peptides were measured using flow cytometry.

**Figure 3 ijms-19-03781-f003:**
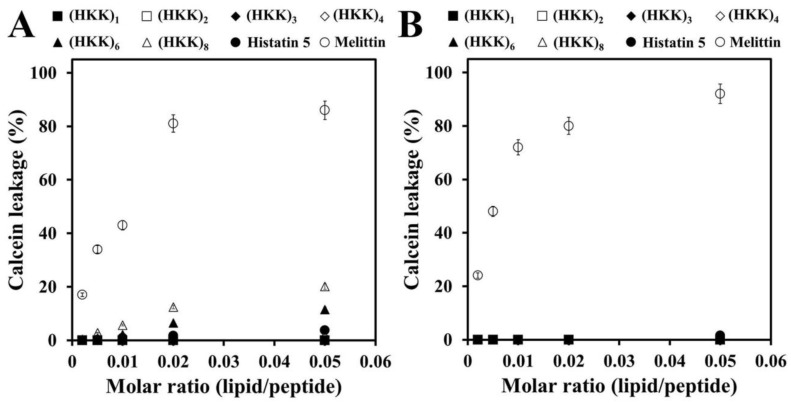
Calcein leakage from calcein-entrapped liposomes was measured by adding peptides at the indicated molar ratios (lipid/peptide). (**A**) PC/PE/PI/ergosterol (5:4:1:2, *w*/*w*/*w*/*w*); (**B**) PC/Ch/SM (1:1:1, *w*/*w*/*w*).

**Figure 4 ijms-19-03781-f004:**
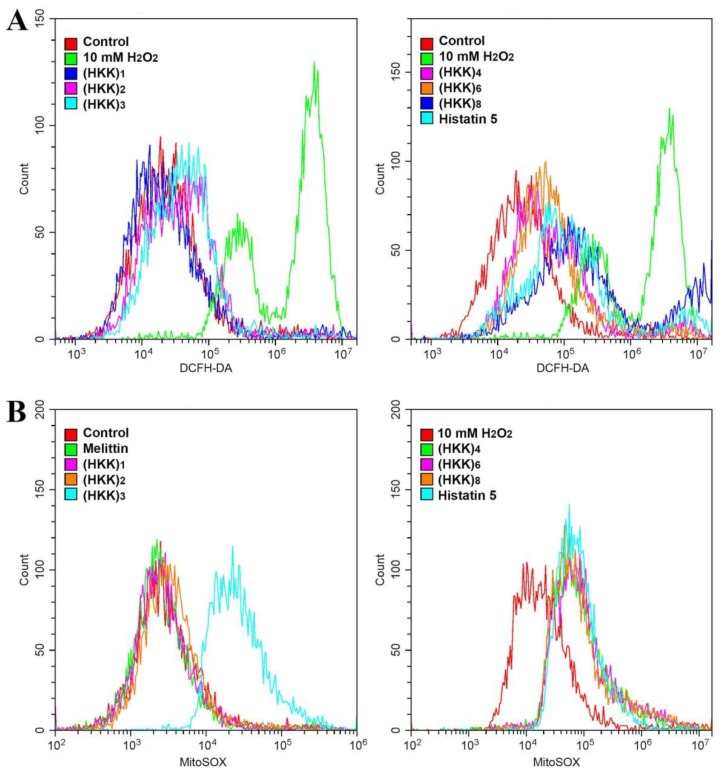
Intracellular ROS (**A**) and mitochondrial superoxide (SOX) generation (**B**) in the presence of peptides in *C. albicans* cells. (**A**) After incubation of the samples with the indicated peptides at their MICs, or with 10 mM H_2_O_2_ for 6 h in *C. albicans* cells, the cells were stained with 100 μM DCFH-DA for 20 min and analyzed using flow cytometry; (**B**) After incubation of the samples with the indicated peptides at their MICs for 6 h in fungal cells, MitoSOX Red was added to the cells and flow cytometry analysis was performed.

**Figure 5 ijms-19-03781-f005:**
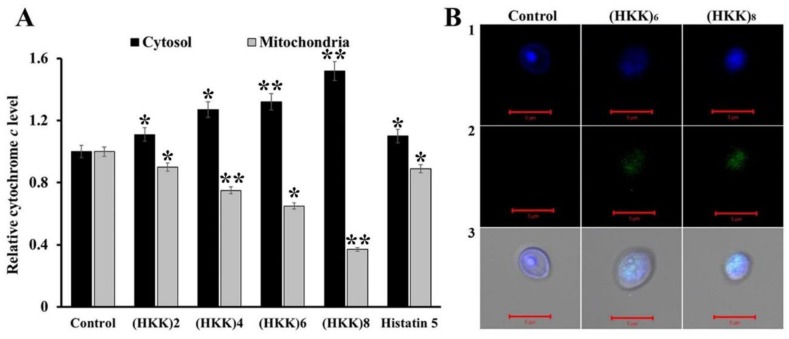
HKK peptide-induced apoptosis in *C. albicans.* (**A**) Release of cytochrome *c* from mitochondria to the cytosol in *C. albicans* was measured at 550 nm. Values represent the mean ± SD (* *p* < 0.05; ** *p* < 0.01), (**B**) CLSM images showing fungi treated with (HKK)_4_ and (HKK)_8_ and double stained with DAPI (1) and TUNEL (2). Merged images are shown in the lower panels (3).

**Figure 6 ijms-19-03781-f006:**
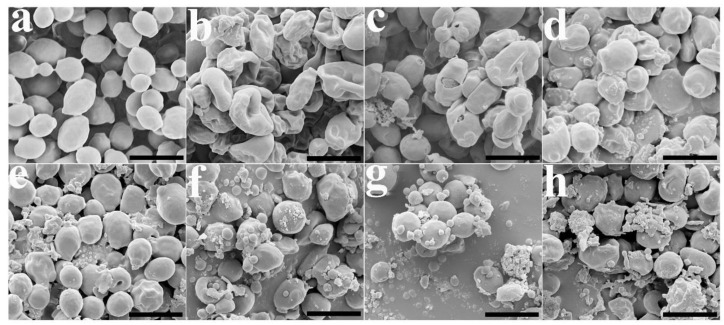
Electron micrographs of *C. albicans* cells treated with peptides for 1 h at their MICs. (**a**) Control (no treatment); (**b**) histatin 5; (**c**) melittin; (**d**) (HKK)_1_; (**e**) (HKK)_2_; (**f**) (HKK)_4_; (**g**) (HKK)_6_; (**h**) (HKK)_8_. Bar is 5 µm.

**Table 1 ijms-19-03781-t001:** Antifungal activity of histidine-lysine-lysine (HKK) peptides against pathogenic fungi.

	MIC (µM) ^a^						
	(HKK)_1_	(HKK)_2_	(HKK)_3_	(HKK)_4_	(HKK)_6_	(HKK)_8_	Melittin
*A. flavus*	256	64	16	8	4	2	8
*C. gloeosporioides*	1024	16	8	2	1	1	16
*C. albicans*	1024	64	64	32	16	8	8
*C. albicans* 14001 ^b^	1024	64	64	32	16	8	8
*C. albicans* 14004 ^b^	1024	64	64	32	16	8	8
*C. albicans* 14007 ^b^	1024	64	64	32	16	8	8
*C. albicans* 14020 ^b^	1024	64	64	32	16	8	8
*C. krusei*	512	64	16	4	4	4	8
*C. parasilopsis*	1024	64	32	32	16	8	8
*F. graminearum*	512	64	32	16	8	4	8
*F. moniliforme*	256	32	16	4	2	2	8
*F. oxysporum*	256	64	32	4	2	2	8
*F. solani*	256	32	16	4	2	1	8
*T. rubrum*	512	64	32	16	8	2	8
*T. virens*	512	32	16	16	8	4	32
*T. viride*	256	64	16	4	2	1	4

^a^ Minimum inhibitory concentrations (MICs) are values for antifungal activity; ^b^ These strains are drug-resistant *C. albicans*.

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
