# Peer review of "Hydrophilic Linear Peptide with Histidine and Lysine Residues as a Key Factor Affecting Antifungal Activity"

_ijms, 2018, doi:10.3390/ijms19123781_

Round 1

Reviewer 1 Report

The submitted manuscript “Hydrophilic Linear Peptide with Histidine and Lysine Residues as a Key Factor Affecting Antifungal Activity” by Park et al have demonstrated novel antifungal peptides with HKK motifs and further investigated their mechanism via confocal microscopy, model membrane, mitochondrial ROS and cell apoptosis. According to their study, HKK peptides can target against phytopathogenic and human-pathogenic fungal strains, which may be used for in vivo study in the future. This is a very interesting study for antifungal agents development, which has been detailed investigated their model of actions. Therefore, I recommend this manuscript to be published on Int J Mol Sci after minor revision.

The authors should address the following comments before publishing.

1.     On page 1 line 28, the authors claimed that “Our results suggested that hydrophobic amino acids were not essential for the antifungal activity of these antimicrobial peptides.” However, they only compared HKK peptides with a single melittin for antifungal activity. According to their study, it cannot be able to get the conclusion that hydrophobic amino acids were not essential for antifungal activity. To support this statement, they should replace one amino acid of the HKK motif to a hydrophobic residue and compare their antifungal activity.

2.     On page 1 line 42, they stated “AMPs have a selective targeting ability in pathogens and multiple modes of action, offering no opportunity for resistance”. However, AMPs are generally having less potential to develop resistance, for example, Kraus D (2006) Curr Top Microbiol Immunol 306, 231. It is inappropriate to claim “offering no opportunity for resistance”. They should rectify it.

Author Response

1.     On page 1 line 28, the authors claimed that “Our results suggested that hydrophobic amino acids were not essential for the antifungal activity of these antimicrobial peptides.” However, they only compared HKK peptides with a single melittin for antifungal activity. According to their study, it cannot be able to get the conclusion that hydrophobic amino acids were not essential for antifungal activity. To support this statement, they should replace one amino acid of the HKK motif to a hydrophobic residue and compare their antifungal activity.

 A) Thank you for your advice. We corrected this sentence to “Our results suggest that peptides composed only of cationic amino acids may be a good candidate as an antifungal agent.”

2.     On page 1 line 42, they stated “AMPs have a selective targeting ability in pathogens and multiple modes of action, offering no opportunity for resistance”. However, AMPs are generally having less potential to develop resistance, for example, Kraus D (2006) Curr Top Microbiol Immunol 306, 231. It is inappropriate to claim “offering no opportunity for resistance”. They should rectify it.

A) We corrected this sentence to “AMPs have a selective targeting ability in pathogens and multiple modes of action, offering less opportunity for development of their resistance”, according to reviwer’s opinion.

Reviewer 2 Report

This article is very interesting report about new antifungal medicine candidates that are hydrophilic repeat cationic peptides. The authors designed these drugs based on the permeability of fungal cell membranes as well as that of mammalian cell membranes, which succeeded in development of harmfulness effective antifungal drugs.

I strongly consider this article provide very useful information about medical treatment of fungal infection diseases.

I have listed some comments about this article to strengthen the theory about some experimental results more clearly.

1) In line 91–94, it was mentioned that (HKK)8 peptide was hemolytic and cytotoxic. Did you analyze the data statistically? Because (HKK)8 peptide looks like similar value in hemolysis (Fig. 1B) and cell survival (Fig. 1C).

2) About Fig.2A, please assign a, b, c and d, respectively. Which panel is FNR 675-labeled (HKK)2, (HKK)4, (HKK)6, and (HKK)8 peptides?

3) In Fig. 2B, could you show the SYTOX Green uptake rate of (HKK)6 and (HKK)8? Both charts of (HKK)6 and (HKK)8 look like almost overlapping with that of Histatin 5. Which SYTOX value is considered “permitted”?

4) Following sentences are complicated (Line 140–144): To further investigate the membrane-permeable effects of peptides in artificial liposomes in which cell wall components, such as membrane proteins and polysaccharides, were absent, 141 phosphatidylcholine (PC)/phosphatidylethanolamine (PE)/phosphatidylinositol (PI)/ergosterol 142 (5:4:1:2, w/w/w/w) and PC/cholesterol (Ch)/sphingomyelin (SM; 1:1:1, w/w/w) vesicles were used as 143 artificial fungal and mammalian model membranes, respectively. 144.

5) At line 140–141, is “in which cell wall components” correct? Isn’t it “cell membrane components”?

6) About Fig. 4A, could you calculate the percentage (or rate) of ROS positive cells?

7) About Fig. 5A, could you analyze this data by statistical methods such as Student’s t-test and ANOVA (analysis of variance)? If you have already done, please mention about the result.  

8) Do you have some opinions why the longer repeating HKK peptides have the capability for anti-fungal effect?

Author Response

1) In line 91–94, it was mentioned that (HKK)8 peptide was hemolytic and cytotoxic. Did you analyze the data statistically? Because (HKK)8 peptide looks like similar value in hemolysis (Fig. 1B) and cell survival (Fig. 1C).

A) We described not “(HKK)8 peptide was hemolytic and cytotoxic” but “(HKK)8 peptide showed very low hemolysis and cytotoxicity” and analyzed statistically all data. So, we corrected to “all HKK peptides were nonhemolytic and noncytotoxic within the tested concentration ranges (Figure 1B and C).”

2) About Fig.2A, please assign a, b, c and d, respectively. Which panel is FNR 675-labeled (HKK)2, (HKK)4, (HKK)6, and (HKK)8 peptides?

A) We added indications of a, b, c, and d in legend of Figure 2.

3) In Fig. 2B, could you show the SYTOX Green uptake rate of (HKK)6 and (HKK)8? Both charts of (HKK)6 and (HKK)8 look like almost overlapping with that of Histatin 5. Which SYTOX value is considered “permitted”?

A) SYTOX Green uptake rates of (HKK)6 and (HKK)8 show in following figures. Uptake percentages of (HKK)6 and (HKK)8 peptides are 19.74 and 31.60%, respectively. Histatin-5 is similar to histogram of control. Uptake rates of (HKK)6 and (HKK)8 were length-dependently increased.

4) Following sentences are complicated (Line 140–144): To further investigate the membrane-permeable effects of peptides in artificial liposomes in which cell wall components, such as membrane proteins and polysaccharides, were absent, 141 phosphatidylcholine (PC)/phosphatidylethanolamine (PE)/phosphatidylinositol (PI)/ergosterol 142 (5:4:1:2, w/w/w/w) and PC/cholesterol (Ch)/sphingomyelin (SM; 1:1:1, w/w/w) vesicles were used as 143 artificial fungal and mammalian model membranes, respectively. 144.

A) We corrected to “To further investigate the membrane-permeable effects of peptides, phosphatidylcholine (PC)/phosphatidylethanolamine (PE)/phosphatidylinositol (PI)/ergosterol (5:4:1:2, w/w/w/w) and PC/cholesterol (Ch)/sphingomyelin (SM; 1:1:1, w/w/w) vesicles were used as artificial fungal and mammalian model membranes, respectively.”

5) At line 140–141, is “in which cell wall components” correct? Isn’t it “cell membrane components”?

A) We deleted “in artificial liposomes in which cell wall components, such as membrane proteins and polysaccharides, were absent,”

6) About Fig. 4A, could you calculate the percentage (or rate) of ROS positive cells?

A) In histogram of flow cytometer, the percentage of ROS positive cells (10 mM H2O2) was 89.64%.

7) About Fig. 5A, could you analyze this data by statistical methods such as Student’s t-test and ANOVA (analysis of variance)? If you have already done, please mention about the result.  

A) We added indication of statistical analysis in figure 5A.

8) Do you have some opinions why the longer repeating HKK peptides have the capability for anti-fungal effect?

A) We suggest that the best antifungal activity of (HKK)8 is due to high cationicity because this causes high binding affinity with fungal cell surface. Lysine has a relatively long aliphatic side chain with a positive charged amine. In particular, hydrophobicity of aliphatic chain can allow strong binding with the hydrophobic tail groups of membrane lipids.
